# Regulation of Replication Stress in Alternative Lengthening of Telomeres by Fanconi Anaemia Protein

**DOI:** 10.3390/genes13020180

**Published:** 2022-01-20

**Authors:** Duda Li, Kailong Hou, Ke Zhang, Shuting Jia

**Affiliations:** Laboratory of Molecular Genetics of Aging & Tumor, Medical School, Kunming University of Science and Technology, Kunming 650224, China; baifufenqi@163.com (D.L.); 20192136012@stu.kust.edu.cn (K.H.); 20202136022@stu.kust.edu.cn (K.Z.)

**Keywords:** replication stress, alternative lengthening of telomeres (ALT), FANCD2, FAMCM, BRCA1/2, RAD51

## Abstract

Fanconi anaemia (FA)-related proteins function in interstrand crosslink (ICL) repair pathways and multiple damage repair pathways. Recent studies have found that FA proteins are involved in the regulation of replication stress (RS) in alternative lengthening of telomeres (ALT). Since ALT cells often exhibit high-frequency ATRX mutations and high levels of telomeric secondary structure, high levels of DNA damage and replicative stress exist in ALT cells. Persistent replication stress is required to maintain the activity of ALT mechanistically, while excessive replication stress causes ALT cell death. FA proteins such as FANCD2 and FANCM are involved in the regulation of this balance by resolving or inhibiting the formation of telomere secondary structures to stabilize stalled replication forks and promote break-induced repair (BIR) to maintain the survival of ALT tumour cells. Therefore, we review the role of FA proteins in replication stress in ALT cells, providing a rationale and direction for the targeted treatment of ALT tumours.

## 1. Introduction

Telomeres are specialized ribonucleoprotein structures in eukaryotic linear chromosome ends, the maintenance of which is critical for the genomic stability and sustained survival of proliferating cells [1,2,3]. Telomeres are progressively eroded with each cell division until the cell cycle checkpoint mediates permanent cell cycle arrest and subsequently triggers replicative senescence and cell death, a process considered through tumour suppression that limits the number of proliferating cell divisions [3,4,5]. In most human cancers, telomerase is reactivated to bypass the replicative crisis during tumorigenesis [5,6]. However, approximately 10–15% of human tumours maintain telomeres by BIR-dependent and homologous recombination (HR)-mediated ALT, which is telomerase-independent [7,8,9]. ALT can be engaged in a wide range of tumor types, and is a common telomere-lengthening mechanism among certain sarcomas (e.g., osteosarcomas and liposarcomas), as well as in subsets of central nervous system tumors, including astrocytomas; however, ALT is relatively rare in the epithelial malignancies [10]. Adopting the ALT pathway confers many unique molecular characteristics to tumors, some of which are commonly used as markers of ALT activity, including (1) high-frequency ATRX mutations, (2) abundant telomeric secondary structures such as R-loops, G4-quadruplexes and telomeric fragility such as extrachromosomal Telomere Repeat (ECTR), (3) active telomeric sister chromatid exchange (T-SCE), (4) ALT-associated PML bodies (APBs) and (5) heterogeneous telomere length [8,9,11,12,13]. The accumulation of R-loops and G-quadruplexes at telomeres may impede the progression of DNA replication, leading to the collapse of replication forks and the formation of one-ended DSBs, which activate BIR-independent telomeric DNA synthesis, consist of DNA end reaction for the formation of 3′ hangout, RAD51/RAD52-dependent invastion, telomere extension mediated by polδ or polη, and Holliday junction resolution named D-loop resolution [14,15,16]. Considering that higher levels of damage are a matter for ALT cells to activate BIR and maintain ALT activity, proteins involved in replication stress are vital to the maintenance of ALT activity.

Fanconi anaemia (FA) is a rare genetic disease characterized by developmental abnormalities, bone marrow failure, and susceptibility to tumours [17]. Studies over the past two decades have identified the crucial function of the FA pathway in maintaining genomic stability [18,19]. The FA pathway contains at least 22 FA proteins and multiple FA-associated proteins, which are divided into upper, middle and downstream parts in the order of action. Upstream is the FA core complex composed of eight FA proteins, including FANCL and other cooperative proteins, which is a multicomponent ubiquitin ligase in essence; Midstream is the ID2 dimer formed by the mono-ubiquitination of the FA core complexes on FANCD2 and FANCI; and Downstream is the DNA repair pathway regulated by FANCI-D2, including FANCR/RAD51-mediated HR, FANCP/SLX4-mediated nuclear acid resection, and replication fork restart mediated by FANCS/BRCA1 [17,20,21]. The FA patient has features of premature aging disorders probably due to the defective DNA damage response, telomere defects, and chronic inflammation [22]. Moreover, the deficiency of the FANC pathway leads to the hallmarks of senescent cells, such as stress-pathway activation (p53/p21, p16, p38/MAPKs), telomere attrition, mitochondrial dysfunction, chromatin alterations, and SASP (senescence associated-secretory phenotype). Some people proposed that FA should be considered also as a cellular senescence-associated disease [23]. Recent studies have found that the FA pathway and associated proteins are involved in the regulation of replication stress in ALT cells and are crucial to the maintenance of ALT activity.

The current study showed that the FA proteins involved in the ALT mechanism mainly have the following:(1)FANCM regulates replication stress in ALT cells in two ways: inhibiting the expression of telomeric repeat-containing RNA (TERRA) to reduce the formation of R-loops and cooperating with the BLM-Topo3α-RMI1/2 (BTR) complex to stabilize and restart the stalled replication fork [24,25,26,27]. (2)FANCD2 is involved in ALT through both ubiquitination-dependent and ubiquitination-independent forms: monoubiquitinated FANCD2 protects nascent DNA from nuclease digestion and stabilizes stalled or regressed forks. Additionally, non-ubiquitylated FANCD2 is anchored to the ALT telomeres by interacting with the COUP-TFII/TR4 complex to promote the loading of the PCNA-POLD3 replication complex on the ALT telomeres [28,29,30,31,32].(3)FA core complex proteins such as FANCJ and FANCA jointly regulate levels of T-SCE by the regulation of HR pathway including BRCA1/FANCS, BRCA2/FANCD1, RAD51/FANCR, and PALB2/FANCN and SLX4/FANCP to maintain ALT telomere length and cell viability [31,33,34,35,36].

Taken together, the extensive involvement of FA proteins in damage induction, T-SCE, and strand elongation in the ALT telomere plays a very important role in the regulation of replication stress in ALT cells. Due to the more recent research elaboration of the role of FANCM and FANCD2 in the maintenance of the ALT mechanism level, this paper will focus on the narrative.

## 2. Source of Replication Stress of the ALT Telomeres

Studies found higher replication levels of stress and damage to telomeres compared to telomerase-dependent tumour cells, which is thought to be an activation signal of the ALT pathway and can promote elongation of ALT telomeres [37]. Numerous studies show that there are multiple sources of replication stress in ALT cells and that these cells are more likely to accumulate replication stress [11,12,13,38,39]. 

With the inability of short telomeres in ALT cells to recruit the Rif2-Rnh201 complex to degrade the R-loop, the accumulation of the R-loop caused stalling or even collapse of the replication fork and the continuous induction of replication stress [40]. Multiple proteins were identified to be involved in the unfolding of the telomere secondary structure, and the chromatin remodelling factor ATRX was one of the most dominant proteins [41]. ATRX mainly mediates the loading of histone H3.3 on telomeres with DAXX, determines telomere epigenetic modification and the degree of loosening, and suppresses telomeric HR [38,42,43,44]. Recent studies have reported that deficiency or low levels of ATRX are popular in the vast majority of ALT cells [45,46,47]. Dysfunction of ATRX promotes CFS expression and TERRA transcription and suppresses the unfolding of the G4 quadruplex and R-loop on ALT telomeres, contributing to the accumulation of telomeric secondary structures that hinder replication fork progression and induce the accumulation of replication stress [48,49,50]. In response to higher levels of replication stress, endonucleases such as BLM in ALT cells are recruited to telomeres to induce stalled replication fork collapse and DNA lesion, activating BIR that promote telomere elongation and cell survival [15,37]. Furthermore, due to the specificity of the telomeric sequence, replication failure needs to restart from the subtelomeric region, with the timing of replication and replication pressure being further increased [51]. Thus, the accumulation of secondary structures on the telomeres in the ALT cells hinders the progression of the replisomes inducing the higher replication stress as compared to the telomerase-positive cells.

Relatively high replication stress can induce the DSB and BIR pathways and promote ALT activity, but excessive replication stress causes DSB accumulation and telomere dysfunction, which induce genomic instability and finally cell death. Regulation of replication stress identifies the activity of the ALT machinery and is critical for ALT cell viability.

## 3. FANCM Unfolds the R-Loop and Stalled Replication Forks to Inhibit Excessive ALT Activity

Although the molecular mechanisms by which the FA pathway is involved in ALT regulation of replication stress are unclear, several FA proteins have been shown to play a role in replication stress. FANCM is an FA core complex component with translocase activity and ATP hydrolase activity that was found to unfold R-loops and stalled replication forks at telomeres in three ways to replicate stress in ALT cells [24,52,53].

First, FANCM is directly localized on the R-loop and Holliday-junction structures by interacting with FAAP24 and MHF1/2, using the branching migration activity of its translocase domain to promote DNA strand migration and replication fork reversal, leading to R-loop expansion and restart of replication forks (Figure 1, Replication Stress, right) [25,26]. Second, FANCM acts as a DNA damage sensor that detects replication DNA damage generated from fork collapse, subsequently activating the assembly of the FA core complex and participating in replication stress in a ubiquitinated FANCD2-dependent manner (Figure 1, Replication Stress, left) [54,55]. Moreover, FANCM interacts directly with the BTR complex to monitor the processing activity of nuclease enzymes, such as the BTR complex, CtIP, Mre11 and related proteins, such as BRCA1, against stalled replication forks and telomeric secondary structures to regulate damage levels, inhibiting excessive ALT activity by limiting BIR activation and replication stress in ALT cells (Figure 1, End Resection) [27,56,57].

Although the recruitment of FANCM to BTR complexes to telomeres is independent of the FA pathway, it is unclear whether other FA core complex components are involved in this process. FANCM deficiency induces significant elevation of telomere and R-loop levels by BLM, resulting in an imbalance between telomere damage and repair, telomere shortening and synthesis, and suppressed ALT cell viability. Regulates of FANCM for replication stress maintain ALT cell survival.

## 4. FANCD2 Stabilizes the Stalled Replication Fork and Participates in BIR to Coordinate Replication Stress in ALT

FANCD2 is a key protein in FA pathway signalling, and normally functional, capped telomeres cannot recruit FANCD2. When ALT telomeres are overshortened and exposed, the FA core complex and ATR complex recognize telomere damage in APBs, recruiting FANCD2 to telomeres involved in replication stress and DNA repair regulation in ubiquitination-dependent and nuclear receptor-dependent ways [28,32]. FANCD2 is mainly involved in replication stress to promote ALT cell survival through two pathways: stabilizing the stalled replication fork and promoting RAD51-mediated telomere synthesis [25,28,29].

Mono-ubibiquitinated FANCD2 is recruited to APBs in response to DNA damage in S phase in an ATR and FA core complex-dependent manner, facilitating intermolecular resolution of the stalled replication fork through translesion DNA synthesis or intramolecular HR, etc., and inhibiting hyperactive ALT by regulating BLM localization to telomeres (Figure 1, Replication Stress and End Resection) [25,28,29,30]. Non-ubiquitinated FANCD2 is anchored to ALT telomeres in G2-phase by the COUP-TFII/TR4 complex interaction to promote the loading of PCNA-POLD3 replication complexes on ALT telomeres by active recruitment of endonuclease MUS81 and BRCA2 to APBs, which is independent of RAD52 (Figure 1, End Resection) [32]. Although FANCD2 has a role in genome-wide replication stress, it mainly suppresses replication stress on ALT telomeres in ALT cells. FANCD2 is primarily dependent on its repair activity and pro-synthetic function in ALT cell replication stress, and FANCD2 deletion leads to an ultrahigh ALT phenotype and telomere loss.

However, the function of monoubiquitinated FANCD2 in the recruitment of nucleases, such as BLM, and the activity of T-SCE in ALT telomeres is controversial. Root et al. found that monoubiquitinated FANCD2 limits the localization of BLM at the telomere to reduce BLM-mediated replication fork collapse in GM847 cells, and depletion of FANCD2 induced elevated levels of telomeric single-stranded DNA and T-SCE [46,51]. However, the data of Fan et al., at U2OS suggest that mono-ubiquitinated FANCD2 contributes to the recruitment of BLM to telomeres, and the depletion of FANCD2 reduces telomeric DNA content and the levels of T-SCE inhibition [50]. Differences in experimental results may arise from differences in protein knockout efficiency, technology, cell state, and cell lines, and subsequent studies should be validated in multiple ALT cell lines and under the same experimental conditions.

It has been reported that FANCD2 directly interacts with TERRA and the R-loop to unwind the R-loop on telomeres and promote recruitment of BLM to resolve G4 quadruplexes. Given the accumulation of telomere secondary structures in ALT cells, it is possible that FANCD2 also regulates ALT activity through this function.

## 5. Regulation of Replication Stress in ALT Cells by Other FA Proteins

Studies have revealed that multiple DNA repair networks converge at ALT telomeres, including BIR, HR, and specialized translesion DNA synthesis (TLS). BRIP1/FANCJ, RAD51 and SLX1/SLX4 from these pathways are also involved in replication stress in ALT cells [58,59].

Certain FA proteins can promote strand invasion and migration in homologous recombination via BIR. In FANCM-deficient ALT cells, PALB2 interacts with BRCA2 and BRCA1 to localize to telomeres, replacing RAD52 and recruiting RAD51 to DNA single strands to form nuclear fibre filaments composed of RAD51, RAD51C, and XRCC2 and telomeres (Figure 1, Strand Invasion of HR) [25]. For ALT telomere dysfunction, PALB2 has different effects on CC formation, which appears in Saos2-dependent PALB2 but not in VA13-dependent ALT cells, while BRCA2 is essential for CC formation in both, suggesting that PALB2 has a BRCA2-independent function in ALT cells [59]. By resolving the stalled replication fork, BRCA2 can directly recruit RNaseH2 to unwind the R-loop on the ALT telomeres (Figure 1, Replication Stress) [39]. The helicase BRIP1/FANCJ can unwind G4 quadruplexes of ALT telomeres same to BLM and cooperate with the TLS complex, which includes PCNA-RAD18 and Polη, to maintain ALT telomere synthesis (Figure 1, Replication Stress) [33,39]. The SLX4-SLX1-ERCC4 complex promotes resolution of the recombination intermediate, resulting in telomere exchange in the absence of telomere extension, which counteracts telomere synthesis induced by the BTR dissolvase complex-mediated processing of recombination intermediates in G2 phase, and SLX4 also solely recruit MUS81 to solve stalled fork in Mitotic DNA Synthes (MIDAS) (Figure 1, MIDAS) [31,34,60].

Moreover, other FA core complex components can significantly affect the mono-ubiquitination of FANCD2 through the FA core complex. For example, FANCA and FANCL play a key role in the mono-ubiquitination of FANCD2, and their deletion significantly inhibits the level of mono-ubiquitination FANCD2 and its recruitment to telomeres, following the rapid shortening of telomeres and inhibition of ALT cell viability [28]. Although FANCC also promotes FANCD2 ubiquitination, its deletion mildly suppresses the level of mono-ubiquitination FANCD2 as well as ALT activity [28].

## 6. Discussion

ALT mechanisms have been identified for nearly two decades, and detailed mechanisms regarding the occurrence and maintenance of ALT remain unclear. Thus far, it has been determined that there are high levels of secondary structure and replication stress in ALT cells, accompanied by telomere loss and high levels of T-SCE, which demonstrates that these replication stresses are required for ALT activation [61,62,63].

This paper summarizes the mechanisms of the occurrence and maintenance of ALT mechanisms and collates with the latest studies on the regulation of ALT mechanisms by FA proteins. Regulation of replication stress is critical to maintaining ALT machinery activity and the survival of ALT cells, but a single mutation in the protein involved in the regulation is insufficient to trigger ALT. There is clearly much more to be learned regarding the molecular behaviour of FA regulating ALT activity at the molecular level and inhibiting CFS expression and R-loop accumulation and the mechanisms by which FA proteins cooperate with different repair factors involved in the regulation of ALT replication stress. It would be interesting to explore the ALT-related functions of the FA protein different from the classic FA pathway. In addition, as mentioned previously, FA deficiency contributed to cell senescence, it is also interesting to identify how FA proteins modulate cells bypass from senescence and activate ALT. ALT is expected to be a target for cancer therapy as a mechanism of telomere maintenance in cancer cells.

Targeted replication stress proteins involved in the maintenance of ALT activity, such as FANCM and FANCD2, abolish the balance between replication stress and damage repair, thus inducing hyperactive ALT and telomere dysfunction, resulting in cell death. Clinical data confirm the feasibility of these ideas. FANCM deficiency is associated with a higher risk of breast cancer and liver cancer, and irreversible lesions are rapidly caused by FANCM deficiency on ALT cells, indicating that short-term inhibition of FANCM can effectively eliminate ALT tumours without secondary effects [64]. HPV-16 E7-induced ALT requires FANCD2 for the maintenance of telomere function, and loss of FANCD2 rapidly leads to ALT telomere dysfunction [30]. Based on inhibiting of the interaction of FANCM with BLM, by targeting the MM2 domain of FANCM with the peptide and the small-molecule drug PIP-199 in vitro showed some discriminatory activity in killing of ALT-cells, compared to telomerase-positive cells. Further development of MM2 mimetics including PIP-199 and peptide would be a valuable strategy for ALT-targeted therapy. Also, several other domains and functions of FANCM may also be amenable to drug development. These structures interacting with FAAP24 and MHF1: MHF2 heterotetramer and ATPase structure, which is also essential for suppression of ALT, could be screened against protein:protein interaction inhibitors [53,61,65], which also applies to FANCD2. With a further understanding of replication stress in ALT cells, more therapeutic strategies will be developed for ALT tumours.

## Figures and Tables

**Figure 1 genes-13-00180-f001:**
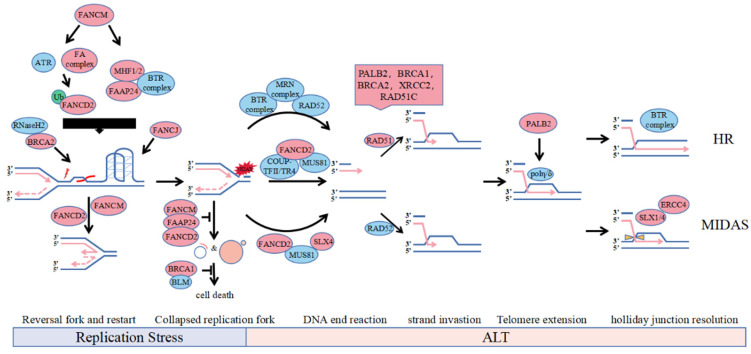
Involvement of FA protein in the ALT pathways. The accumulation of telomere secondary structure such as R-loops and G-quadruplexes may lead to replication fork stalling. During replication stress, the slow or stalled replication fork can undergo fork reversal and restart by FANCD2 and FANCM. BRCA2 can recruit RNaseH2 to unwind R-loop and FANCJ can solely resolve G4 on ALT telomeres. Depletion of FANCM or FANCD2, which is synthetically lethal with BRCA1 or BLM, reduces the replication efficiency of telomeric DNA and increases formation of ECTR and micronuclei. However, unresolved replication stress leads to fork collapse and genome instability catalyzed by nucleases including SLX4 and MUS81 recruited by FANCD2, which provides direct DNA lesion or DSB for substrates to induce BIR mediated ALT. Recruitment on telomere of HR factor for strand invastion in S phase, polη stabilized by PALB2 for telomere extension and SLX1/4 for recombination intermediate dissolution in mitotic phase contribute to ALT-mediated recombination and telomere synthesis.

## Data Availability

Not applicable.

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
