# Peer review of "Regulation of Replication Stress in Alternative Lengthening of Telomeres by Fanconi Anaemia Protein"

_genes, 2022, doi:10.3390/genes13020180_

Round 1

Reviewer 1 Report

The present review summarizes the currently known implications of FA proteins in the regulation of replication stress in ATL cells. The research covers with high detail the mechanisms that promote this replication stress in ALT cells, and how FANCM, FANCD2 and other FA proteins interact with different types of DNA structures involved in ALT.

General concept comments:

  • It could improve the quality of this review to highlight even more the implication of this ALT cells in different tumors. Which characteristics confer to tumor cells this ALT activity?

  • It could help to make the review more comprehensive to the reader a brief explanation of ALT process, before introducing the FA proteins effects. Also, if possible, a figure would ease even more the reading.

  • Is there any study stablishing a direct connection between any FA proteins and cell senesce/cell death? If it exists, it would be interesting to mention it.

Specific comments:

  • There is an excessive citation in the introduction, 63 cites in the first 72 lines. It would be preferable to reduce this quantity, maintaining only the essential papers and most recent original works.

  • Please change the following expression: “Numerous studies in this study show…” (line 81-82).

  • As it does appear only once, please change the “MIDAS” abbreviation (line 190).

  • Please correct the format of the sentence located between lines 204-205.

  • I would delete the sentence “We expect that the next decade will be an exciting time for 236 both basic and translational research on ALT.” (lines 236-237), because of its subjective content.

  • There are some changes in the style of the bibliography, mostly in bold type letters. For example, lines 261, 264 or 267. Please review the citation style of the paper.

Reviewer 2 Report

The study is relatively well designed and executed. However, some major revisions should be undertaken prior to the publication of this manuscript.

Firstly, there are some issues with the English (syntax, grammar, etc.).

The authors do not cite relevant literature and thus seem to lack knowledge on prior art/state-of-the-art.
If there are any drugs targeting FA ( antibody, peptide, small molecule or etc) in research or clinical trials? If so, pls indicate them in a table. 

What is the suggestion of the authors for future studies?

Round 2

Reviewer 2 Report

I am satisfied with the authors' responses to my comments and suggest
publication of the manuscript to the Genes.